# Long Non-Coding RNAs as Epigenetic Regulators of Immune Checkpoints in Cancer Immunity

**DOI:** 10.3390/cancers15010184

**Published:** 2022-12-28

**Authors:** Wiam Saadi, Ahlam Fatmi, Federico V. Pallardó, José Luis García-Giménez, Salvador Mena-Molla

**Affiliations:** 1Department of Biology, Faculty of Nature, Life and Earth Sciences, University of Djillali Bounaama, Khemis Miliana 44225, Algeria; 2INCLIVA Health Research Institute, INCLIVA, 46010 Valencia, Spain; 3Center for Biomedical Network Research on Rare Diseases (CIBERER), Institute of Health Carlos III, 46010 Valencia, Spain; 4Department of Physiology, Faculty of Medicine and Dentistry, University of Valencia, 46010 Valencia, Spain

**Keywords:** cancer immunity, epigenetic regulation, immune checkpoint, long non-coding RNAs, circular RNA

## Abstract

**Simple Summary:**

Many non-coding RNAs are deregulated in cancer. In one of the most intriguing of their various roles, they control immune regulatory proteins, allowing tumor cells to evade the body’s own defenses. In this work, we describe the mechanisms by which certain ncRNAs help tumor cells escape the immune response. In-depth insight into these mechanisms is required to further develop therapeutic strategies targeting immune checkpoints and improve the performance of cancer immunotherapy.

**Abstract:**

In recent years, cancer treatment has undergone significant changes, predominantly in the shift towards immunotherapeutic strategies using immune checkpoint inhibitors. Despite the clinical efficacy of many of these inhibitors, the overall response rate remains modest, and immunotherapies for many cancers have proved ineffective, highlighting the importance of knowing the tumor microenvironment and heterogeneity of each malignancy in patients. Long non-coding RNAs (lncRNAs) have attracted increasing attention for their ability to control various biological processes by targeting different molecular pathways. Some lncRNAs have a regulatory role in immune checkpoints, suggesting they might be utilized as a target for immune checkpoint treatment. The focus of this review is to describe relevant lncRNAs and their targets and functions to understand key regulatory mechanisms that may contribute in regulating immune checkpoints. We also provide the state of the art on super-enhancers lncRNAs (selncRNAs) and circular RNAs (circRNAs), which have recently been reported as modulators of immune checkpoint molecules within the framework of human cancer. Other feasible mechanisms of interaction between lncRNAs and immune checkpoints are also reported, along with the use of miRNAs and circRNAs, in generating new tumor immune microenvironments, which can further help avoid tumor evasion.

## 1. Introduction

Cancer is one of the most complex multifactorial human diseases. In 2020, 10 million people are estimated to have died from cancer [1], reflecting an ongoing need to discover new and efficacious drug combination therapies.

Cellular neoplastic transformation is a dynamic process underpinned by numerous molecular and cellular dysfunctions, including accumulative genetic alterations (mutations, loss of heterozygosity, deletions, insertions, aneuploidy, etc.) and/or epigenetic abnormalities [2,3]. Epigenetics involves heritable changes in gene expression that are not mediated by changes in DNA sequence [4]. Since epigenetic pathways are important for organism growth and tissue-specific homeostasis, their alteration can result in pathological states [5]. An important role has been reported for epigenetic alterations in controlling the expression of tumor-associated genes, such as tumor suppressors, oncogenes, and inhibitory cytokines, as well as in encoding immunological checkpoint molecules, which result in uncontrolled tumor growth and defective anticancer immunity. This can ultimately lead to tumor development, progression, metastasis, treatment resistance, and immune evasion [5,6]. The key processes responsible for epigenetic mutations (epimutations) include aberrant DNA methylation, abnormal histone modifications, and altered expression of various non-coding RNAs (ncRNAs), such as small (sncRNAs) and long non-coding RNAs (lncRNAs), including circular RNAs (circRNAs) [7]. These changes can be reversed using drugs targeting epigenetic marks associated with cancer to restore immune surveillance and homeostasis. These drugs, known as epigenetic drugs or “epi-drugs”, target DNA methyltransferases (DNMT) and DNA demethylases, histone deacetylases [HDAC] [8], and even ncRNAs [9]. Many recent studies show that combining epi-drugs with immunotherapeutic treatments might improve anticancer immune responses [10].

Despite the potential role of immunotherapy in cancer treatment, the tumor immune escape system remains the most difficult challenge to overcome. This system is the last stage of the cancer immunoediting process, in which tumor cells develop the ability to avoid immune system identification, employing a variety of strategies including decreased immunogenicity, dysregulation of cell metabolism and cytokine production, induction of aberration in immunosuppressive cells, and overexpression of immunological checkpoints [11]. Immune checkpoints (ICs) are specific membrane molecules used by the immune system to distinguish between normal and abnormal cells in the body [12]. IC dysregulation is one of the hallmarks of malignancies and autoimmune disease. As noted above, ICs and their ligands are selectively increased in many tumor cells, depleting the proliferative capacity of T cells and their capacity to produce cytokines, thus resulting in immune tolerance and tumor evasion [13,14,15].

Cancer immunotherapies based on IC inhibitors (ICIs) or chimeric antigen receptor T (CAR-T) cell therapy have opened new therapeutic avenues in the treatment of cancer due to their remarkable clinical efficacy in some patients. ICIs are antibodies targeted to ICs, which re-activate the host immune defenses against tumor cells, and are currently used in first or second treatment lines for several cancer types [16].

Despite these promising advances, a large percentage of patients (60–80% depending on the line of treatment) fail to respond to ICIs or develop resistance [17]. Since ICIs activate the immune system, they often show adverse immunotherapy-related effects [16,18]. This highlights a need for increased insight into tumor immune evasion in order to identify new potential therapeutic targets that will improve immunotherapy efficiency. In this regard, many studies have shown that epigenetic alterations, such as deregulation of histone post-translational modifications, abnormal DNA methylation, and altered ncRNA expression, can directly impact IC regulation [19].

Research into the role of epigenetic mechanisms associated with cancer immune evasion has focused mainly on histone modifications, microRNA (miRNA) inhibition, and DNA methylation [20,21]. Currently, other ncRNAs implicated in IC regulation are slowly emerging as attractive anticancer treatment targets. Here, we provide an overview of current understanding of the role of lncRNAs in modulating IC expression, with an emphasis on the roles these mechanisms play in cancer initiation and progression, which to our best knowledge has not been systematically summarized. Moreover, we aim to pinpoint the lncRNA candidates most likely to have a therapeutic impact in the near future, thus contributing to the development of novel cancer treatment strategies.

## 2. Immune Checkpoint Molecules and Cancer Progression

The immune system is a combination of tissues, cells, and signaling molecules that work together to defend the body by identifying and destroying foreign cells while saving healthy ones and maintaining self-tolerance [22]. 

Our system constantly generates neoplastic cells. During tumorigenesis, tumor cells express distinct tumor-specific markers that are recognized by the immune system cells and molecules of innate and adaptive immunity, leading to the eradication of these neoplastic cells, a process named immunosurveillance [23]. However, these cells can escape the immune system in a variety of ways, leading to tumor formation and evasion from immune control, one of the hallmarks of cancer as defined by Hanahan and Weinberg [3,24].

Immune system pressure forces tumor cells to edit the surrounding immune system in order to survive, a process known as cancer immunoediting. Cancer immunoediting involves a first elimination phase in which the immune system recognizes and kills cancer cells. The next phase is an equilibrium stage, with coexistence between tumor cells and immune cells. In the final phase the tumor escapes from immune control [25]. 

In the first step of the cancer–immunity cycle, cancer antigens are released by transformation of normal cells to cancer cells (oncogenesis) during apoptosis or cell death. These cancer antigens are captured by dendritic cells (DCs), which then mature and migrate to lymph nodes (Step 2) to process and present the captured cancer antigen to the major histocompatibility complex class (MHC) I molecule, resulting in T cell priming (Step 3). The activated effector T cells migrate to the tumor (Step 4) and infiltrate the tumor tissue (Step 5). Within the tumor, the T cells recognize cancer cells (Step 6) and injure them (Step 7), releasing additional antigens and restarting the cycle [26] (Figure 1). This cancer–immunity cycle is disrupted in cancer due to immunoediting, a process by which immune function can shape tumor immunogenicity.

The capacity to distinguish between normal and tumor cells in the body is a hallmark of the immune response. To prevent our immune system from destroying our own cells, steps 3 and 6 of the cancer–immunity cycle function as ICs, where T cell activation is controlled. Activation of lymphocytes requires not only specific antigen recognition by lymphocytes, but also an additional co=stimulatory signal involving ICs [27]. These molecules may either boost or reduce immune system signals [28]. Stimulatory checkpoint mechanisms organize the formation of naïve T cells, as well as regulatory, memory, and effector T cells. Inhibitory checkpoint mechanisms, on the other hand, limit the threshold and duration of T cell activation, affecting inflammation, tolerance, and homeostasis resolution. These molecules are engaged on the surface of both T cells and cancer cells, functioning as receptors and ligands. The best known and characterized checkpoint proteins and their ligands are as follows: T-lymphocyte-associated protein 4 (CTLA-4; CD152), CD80 (B7-1) and CD86 (B7-2); programmed cell death protein 1 (PD-1; CD279), PD-L1 (PD-1 ligand 1; CD274; B7-H1) and PD-L2 (CD273; B7-DC); lymphocyte-activation gene 3 (LAG3; CD233), MHC class II ligation; T cell immunoglobulin and mucin-domain containing-3 (TIM-3; CD366), galectin-9 or phosphatidyl serine or carcinoembryonic antigen-related cell adhesion molecule (CEACAM-1); B- and T-lymphocyte attenuator (BTLA; CD272), Herpes Virus Entry Mediator (HVEM); and T cell immunoreceptor with Ig and ITIM domains (TIGIT), CD155 (Table 1).

ICs and their ligands are commonly upregulated in the tumor microenvironment. These ICs are inhibitory receptors expressed in B and T lymphocytes; antigen-presenting cells (APC) like DCs or macrophages; innate lymphoid cells, which include natural killer (NK) cells; and some cancer cells. Their ligands can be expressed on tumor cells as well as on a wide variety of immune (i.e., T and B cells, NK cells, monocytes, macrophages, and DC), and non-immune cells (i.e., epithelial, vascular endothelial, and many other cells) [40,41,42].

CTLA-4 was the first IC identified as a target to enhance T cell immunity in tumor-bearing mice and was also the first to be targeted therapeutically. Anti-CTLA-4 blockade with ipilimumab was the first treatment to prolong overall survival (OS) in patients with advanced melanoma (Figure 2), yet checkpoint therapy against PD-1 has proven more effective in several other types of cancer [43].

CTLA-4 is an inhibitory receptor that competes with the activator receptor CD28 for binding on CD80 or CD86 molecules expressed on APC cells. However, T cell receptor (TCR) binding to MHC is also required for APC activation. When the TCR is strongly stimulated, CTLA-4 levels increase, which results in augmented CTLA-4/CD80 or CD86 binding, which in turn reduces IL-2 synthesis, T cell proliferation, and survival [44,45]. 

PD-1 can interact with either PDL-1 or PDL-2. Ligand binding to PD-1 activates two intracellular tyrosine-based structural motifs: immunoreceptor tyrosine-based inhibitory motif (ITIM) and an immunoreceptor tyrosine-based switch motif (ITSM), which recruit and activate a Src homology region 2 domain-containing phosphatase-2 (SHP-2), resulting in dephosphorylation and downregulation of downstream positive signaling pathways triggered by TCR-MHC– CD28-CD80/CD86 interaction, such as inhibition of ZAP70 and the PI3K/AKT and RAS signaling pathways. PD-1 activation decreases the expression of transcription factors (TFs) that promote T cell activation, proliferation, and survival, such as activator protein 1 (AP-1), nuclear factor of activated T cells (NFAT), and nuclear factor-B (NF-B), leading the cell to an exhausted state, reducing its functions and blocking interferon γ (IFN-γ) production and cytotoxicity [45,46]. On other cells, such as regulatory T cells (Tregs), PD-1 decreases the magnitude of the immune response in T cells that are already involved in an effector T cell response [44].

Targeting of PD-1 and CTLA-4 formed the basis of the first generation of checkpoints, with the next generation including LAG-3, TIM-3, BTLA, and TIGIT. Like PD-1 and CTLA, these ligands regulate immune suppression by inhibiting T cell activation and cytokine secretion. These data are shown in more detail in Table 1 [47].

Increased IC expression is well-documented as correlating with worse prognosis and a shorter overall survival ratio (OS) in different cancer types. Increased CTLA-4 expression in circulating CD4+ T cells from colorectal cancer patients was found to be positively correlated with tumor-node-metastasis (TNM) staging [48]. Increased PD-L1 and LAG-3 expression in tumor tissues from triple-negative breast cancer patients receiving adjuvant treatment has been linked to poor prognosis [49]. Zhang et al. reported that TNM staging, lymph metastasis, and shorter OS were all strongly related with TIM-3 expression in colorectal tumor tissues [50] and hepatocellular carcinoma [51,52]. B- and T-lymphocyte attenuator (BTLA) is a receptors expressed on the surface of T cells, B cells, DC, and myeloid cells with significantly higher expression in cancer patients compared to healthy controls [40]. TIGIT is expressed in T cells and NK cells. When TIGIT interacts with its primary ligand CD155 on DC or cancer cell membranes or with CD112 expressed by tumor cells and APC in the tumor microenvironment, it induces a decrease in the proinflammatory cytokine secretion, proliferation, and killing function of immune cells [53,54]. 

Interestingly, ICs molecules also have soluble isoforms, formed either by proteolytic cleavage of the membrane-bound form or by alternatively spliced mRNA translation and subsequent secretion of the protein product by immune cells [55]. There is scientific evidence that these entities can participate in immune regulation through interactions between soluble form receptors and full-length ligands or between soluble ligands and full-length receptors [55]. Moreover, changes in IC plasma levels can also contribute to cancer development and prognosis, and even affect the effectiveness of treatments (Table 2).

These IC molecules are involved in the development of cancer and poor prognosis and have been suggested as biomarkers and therapeutic targets [55]. Therefore, IC therapy is currently considered a cornerstone of cancer treatment and has led to the creation of ICIs, monoclonal antibodies that target ICs, and their ligands for cancer treatment (Figure 2) [16,65,66].

## 3. Long Non-Coding RNAs: An Overview

Non-coding RNAs (ncRNAs), thought of as “junk” DNA for decades, are a large and diverse group of RNAs that make up more than 70% of the human genome and comprise 98% of the transcriptome but are mostly not translated into proteins [67]. Instead of coding for a specific protein, these molecules can target and control the expression of other biomolecules [68]. Based on their size, ncRNAs are broadly categorized into two groups: (1) sncRNAs, which are shorter than 200 nucleotides, including miRNAs, PIWI-interacting RNAs (piRNAs), transfer RNAs (tRNAs), ribosomal RNAs (rRNA), small nuclear RNAs (snRNAs), small nucleolar RNAs (snoRNAs), and small interfering RNAs (siRNAs); and (2) lncRNAs, which as the name suggests are long non-coding RNA transcripts, with more than 200 nucleotides [69,70]. In recent years, great progress has been made in our understanding of how ncRNAs govern and regulate processes in cells and organisms, and how these mechanisms contribute to the progression of many diseases, including cancer, when they go wrong [71]. Studies have reported the involvement of ncRNA deregulation in multiple cancer types, and they appear to play a role in multiple cancer processes, including cancer initiation, progression, metastasis, and drug resistance [72]. For these reasons, ncRNAs could also be targets for cancer treatment, with the potential even to be used as novel prognostic and diagnostic markers [73]. Despite being a rapidly developing field of biomedical study, the significance of ncRNAs in several human disorders remains unknown [74]. In cancer research, the greatest interest has been focused on miRNAs, a highly conserved single-stranded RNA that belongs to the sncRNA class (22–25 nucleotides in length) [75].

Myriad lncRNAs have been reported in humans to date, including 19,933 lncRNA genes and 57,936 lncRNA loci transcripts (GENCODE v42; www.gencodegenes.org, accessed on 12 July 2022). LncRNAs are the most heterogeneous and common type of ncRNA (although the majority of lncRNAs transcripts share similarities with mRNA; they are transcribed by RNA polymerase II, spliced, capped, and polyadenylated) [76,77]. These multiple properties of lncRNAs make it difficult to accurately divide them into different groups. The appropriate technique to stratify lncRNAs into groups is therefore still under debate, as classification can infer information on areas such as structure, chromosomal location, regulatory role, and biological function.

Figure 3 describes some of the most accepted (not mutually exclusive) current classifications. Based on their genomic location regarding protein-coding genes (PCG), lncRNAs can be classified as follows: sense lncRNAs when they are on the same strand of PCG, antisense lncRNAs on the opposite strand of PCG, intronic lncRNAs if located entirely in intronic regions of a PCG, intergenic lncRNAs (lincRNAs) when they are located in intermediated regions of PCG, and bidirectional lncRNAs when they are transcribed from the opposite strand of PCG within 1 kb of the promoter region of the PCG [70]. LncRNAs in an antisense direction of the overlapping PCG are called natural antisense transcripts [78], and they frequently control the sense strand [78].

LncRNAs can also be categorized into the following groups according to their genomic location and association with specific DNA regulatory regions [79]: (1) Pseudogene-derived lncRNAs, which are transcribed from superfluous copies of functional genes that have lost their coding potential during evolution [80]. (2) T-UCR lncRNA (T-UCRs or TUCRNA), which are transcribed from ultraconserved regions [81]. (3) Enhancer lncRNAs (elncRNAs), which are transcribed from genomic regions enriched with TF-binding sites [82]. If lncRNAs are located in large clusters of enhancers, they are called super-enhancer lncRNAs (selncRNAs). SelncRNAs recruit cofactor, RNA polymerase II, and a mediator to form and maintain the chromatin loop SE and promoter region, which controls the transcription of target genes [83]. (4) Promoter-associated lncRNAs (pancRNAs or pncRNAs), the most abundant single-copy lncRNA biotype, are codified in the PCG promoter [84]. (5) Finally, UTR-associated lncRNAs are located in the 3′ untranslated regions (UTR) of a PCG [85]. 

Structurally, lncRNAs are divided into linear RNAs and circular RNAs (circRNAs). Linear lncRNAs can originate from their own loci or from alternative splicing of mRNAs. CircRNAs are single-stranded closed ncRNA molecules, generated by reverse splicing of mRNAs, and are more resistant to the enzymatic activity of RNase. Specifically, circRNAs are generated from exons of PCG [86] or introns [87] by a back-splicing process catalyzed by a spliceosome, which binds a downstream splice donor site (5′ splice site) to an upstream acceptor splice site (3′ splice site) [88,89]. We recently described in detail the synthesis mechanisms of circRNAs and their potential use as diagnostic markers and therapeutic potential [90]. Currently, circRNAs are mainly classified into five types: exonic circRNAs (EciRNAs), exon-intron circRNAs (EIciRNAs), intronic RNAs (CiRNAs), fusion circRNAs (f-circRNAs), and read-through circRNAs (rt-circRNAs) [91]. 

Another classification is based on their mechanism of action. We have established three hierarchical functional levels, modifying the proposal of Cheng et al. [92]. At the first level, lncRNAs are classified based on their ability to form complexes with DNA, RNA, or proteins through direct binding or through recruitment by specific structures generated by RNA folding [93]. Based on their molecular functions, lncRNAs can be categorized into four archetypes: (1) signals, by activating or repressing genes; (2) decoys, by blocking other regulatory RNAs or proteins from accessing their target, i.e., competing endogenous RNAs (ceRNAs) for miRNAs; (3) guides, by forming a large networks of ribonucleoprotein (RNP) complexes with multiple chromatin regulators and then recruiting them to specific targets; and (4) scaffolds, by acting as pivotal platforms for assembling distinct effector molecules [94]. Finally, due to their complexity and flexibility, lncRNAs perform a variety of regulatory functions and may be categorized as follows [95,96]: (1) transcriptional regulators interacting with TF or the transcription machinery, via formation of chromatin loops or activating enhancer elements embedded in their loci; (2) post-transcriptional regulators through binding to proteins, target mRNAs, or miRNAs to modulate mRNA splicing, turnover, or signaling pathways; (3) regulators of cellular organelles through interaction with proteins that participate in the synthesis and homeostasis of organelles such as mitochondria or exosomes; (4) structural regulators, as architectural RNAs (arcRNAs) that serve as scaffold for components of nuclear bodies; (5) genome integrity and chromatin state regulators via direct interaction with chromatin or through interaction with chromatin modifiers. 

Another proposed method of lncRNA classification is based on the locus on which they act. According to their effect they can be described as: (1) cis-acting lncRNAs (cisRNAs), when they regulate the adjacent loci from which the lncRNA is transcribed; (2) trans-acting lncRNAs (transRNAs), when they regulate loci on other chromosomes in the cell; and (3) secreted lncRNAs, when they are exported to act in other cells like exosomal lncRNAs [97].

## 4. LncRNAs Involved in Epigenetic Regulation of Immune Checkpoints in Malignancies 

In recent years, a large number of lncRNAs related to cancer have been identified. However, much more remains to be studied to gain deeper insight into their mechanisms of action. In this section, we describe a selection of lncRNAs with at least one well-characterized mechanism associated with immune evasion. Figure 4 and Table 3 provide a summary of these lncRNAs. 

Tang and colleagues discovered that the lncRNA actin filament-associated protein 1 antisense RNA1 (AFAP1-AS1) was overexpressed in nasopharyngeal carcinoma cells, with levels positively correlated with PD-1 expression [98]. High levels of PD-1 or AFAP1-AS1 in tumor-infiltrating lymphocytes are linked to poor patient prognosis, whereas simultaneous expression of both AFAP1-AS1 and PD-1 is linked to the lowest survival rate [98]. However, it is unknown whether AFAP1-AS1 can enhance PD-1 expression on its own. More research is therefore needed to clarify this intricate mechanism. In the NPInter integrated database of ncRNA interactions (http://bigdata.ibp.ac.cn/npinter4, accessed on 12 July 2022), a possible RNA–RNA interaction was found between AFAP1-AS1 and phosphoprotein associated with glycosphingolipid-enriched microdomains 1 (PAG), a downstream effector of PD-1 that contributes to the inhibitory function of PD-1 in T cells. It is noteworthy that patients unresponsive to PD-1 blockade therapies show high PAG expression [99]. Therefore, a possible AFAP-AS1 mechanism of action could be related to the PD-1/PAG axis.

LncRNA C5orf64 was found to be positively correlated with PD-1, PD-L1, and CTLA-4 [100]. The authors note that C5orf64 might regulate PD-L1 expression. There are distinct transcriptional control mechanisms of PD-L1 in cancer, including signaling pathways such as RAS/RAF/MEK/MAPK-ERK, PI3K/PTEN (a negative regulator of PI3K)/Akt/mTOR, and JAK/STAT. Different TFs, such as c-Jun, HIFs, NF-κB, STAT1, STAT3, and ZEB, can shuttle into the nucleus, binding to the *CD274* (also called PD-L1 or B7-H1) promoter to induce its expression [101,102]. C5orf64 might regulate PD-L1 expression by activating the epidermal growth factor (EGF) receptor (EGFR), an upstream effector of the RAS/RAF/MEK/MAPK-ERK signaling pathway. The role of C5orf64 on this pathway involves the miRNA-150/EREG axis [100], which adds a new component of complexity to this mechanism. 

Despite the lack of extensive research on the C5orf64/miRNA-150/EREG axis, it has been confirmed that miR-150 is associated with higher immune cell infiltration (lymphocytes and APC cells), higher IC expression (PD-1, CTLA4, TIGIT, BTLA or LAG3), and higher survival rates [103]. MiR-150 also targets EREG or epiregulin (both mi-150-5p [104], and mir-150-3p (TargetScan; https://www.targetscan.org, accessed on 12 July 2022)), a member of the EGF family of peptide growth factors that is deregulated in several forms of cancer. EREG is an EGFR ligand that has been associated with immune suppression by activating Tregs, decreasing levels of the T cell chemoattractant CCL27 [105], and increasing PD-L1 expression [106]. 

LncRNA CASC11, which has been found upregulated in several cancers, is located upstream of MYC and has been described as an activator of the Wnt/β-catenin and the PI3K/AKT signaling pathway [107]. One possible mechanism consists of binding CASC11 to the RNA-binding protein EIF4A3: the complex increases the stability of the E2F Transcription Factor 1 (E2F1) mRNA, activating the NF-κB and PI3K/AKT/mTOR pathways, which finally control PD-L1 expression [108]. In fact, controlling the PI3K/Akt/mTOR pathway should be addressed via several strategies, such as targeting lncRNAs directly involved in the upregulation of these pathways, since the use of pharmacological inhibitors of PI3K/AKT/mTOR has been proven to regulate the expression of immune checkpoint ligands and interfere with immune evasion [109]. 

Despite its still unknown role in cancer, increased expression of lncRNA cat eye syndrome chromosomal region, candidate 7 (CECR7) is associated with poor prognosis in hepatocellular carcinoma [110] and higher OS in diabetic pancreatic cancer [111] and metastasized colorectal cancer [112]. Regarding its mechanism of action on ICs, it has been reported that CECR7 regulates the immune response by targeting miR-429, which regulates CTLA-4 expression [111]. 

miR-429 is part of the miR-200 family consisting of miR-200a, miR-200b, miR-200c, miR-141, and miR-429. This family targets ZEB1/2, which plays an important role in immunosuppressive checkpoint expression. For example, ZEB1 can directly or indirectly upregulate the expression of PD-L1 [113,114]. 

The lncRNA metastasis-associated lung adenocarcinoma transcript 1 (MALAT1) is a known inhibitor of the miR-200 family, but can also bind to miR-195 to upregulate the PD-L1 expression (miR-195 binds to 3-UTR of miRNA PD-L1 [115]), promoting migration and immune escape through regulating CD8+ T cell proliferation and apoptosis in diffuse large B cell lymphoma patients [116]. Regarding its action on the miR-200 family, MALAT1 inhibits miR-200c, which negatively regulates ZEB1 [117] and also inhibits miR-200a, which directly targets the PD-L1 mRNA [116]. MALAT1 can therefore control the expression of PD-L1 at different levels.

The lncRNA SNHG14 downregulates miR-5590-3p through a sponging mechanism. ZEB1 is a miR-5590-3p downstream target, so its overexpression promotes cancer cell migration and metastasis by inducing the PD-1/PD-L1 axis. ZEB1/PD-1 also enables immune evasion in cancer cells [118]. The SNHG14-miR-5590-3p-ZEB1 axis positively regulates PD-L1 [119]. Thus, understanding this key axis in PD-L1 upregulation may contribute towards creating new therapeutic strategies, since targeting the PD-L1/PD-1 pathway has consistently shown significant and promising therapeutic efficacy in patients with advanced cancers [120]. Interestingly, miR-5590-3p has been shown to inhibit proliferation and metastasis of renal cancer cells [121] and also tumor growth in gastric cancer through the DDX5/AKT/m-TOR pathway [122].

In this regard, regulation of miR-5590-3p expression and its interaction with SNHG14 warrants further investigation, as this mechanism may set the basis for new therapeutic strategy by affecting the function of the PD-L1/PD-1 immune checkpoint. 

EMX2OS is an elncRNA located in the enhancer region of the *EMX2* gene. It acts as a miR-654 sponge, activating AKT3 and resulting in PD-L1 overexpression [123]. High EMX2OS levels are associated with low OS in adrenocortical carcinoma, cervical squamous cell carcinoma, endocervical adenocarcinoma, kidney renal clear cell carcinoma, stomach adenocarcinoma, uveal melanoma, and gastric cancer [124]. It would therefore be very useful to measure EMX2OS levels in ICI-resistant patients and tailor strategies to these EMS2OS levels. 

Downstream inhibition is also a plausible mechanism in IC regulation. For example, miR-142-5p overexpression inhibits pancreatic cancer growth by targeting PD-L1 expression on tumor cells, resulting in an increase in IFN- and TNF-, an increase in CD4+ and CD8+ T lymphocytes, and a decrease in PD-1 T lymphocytes [125]. However, lncRNA FGD5-AS1 functions as a sponge of miR-142, reducing its regulatory effect on PD-L1 expression [126]. Furthermore, FGD5-AS1 can act as a miR-454-3p sponge, increasing ZEB1 levels and PD-L1 expression [127]. This provides other example of the intricate mechanisms underlying overall IC regulation involving lncRNAs and miRNAs.

PD-L1 is linked to the GATA-binding protein 3 antisense RNA 1 lncRNA (GATA3-AS1) in triple-negative breast cancer. This lncRNA sequesters miR-676-5p, a negative regulator of COP9 signalosome 5 (CSN5; also known as COPS5). Therefore, GATA3-AS1 increases expression of COPS5, which encodes a deubiquitinating enzyme that removes ubiquitins and stabilizes PD-L1 [128]. There was an increase in the percentage of CD8+ T cells when co-cultured with silenced GATA3–AS1 in the breast cancer cell lines MDA-MB-231 and HCC1937 [128].

There are 18 lncRNAs identified in the clusters encoded by homeobox (HOX) genes, which are located in four clusters: the HOXA region, consisting of HOTTIP, HOTAIRM1, HOXA-AS2, HOXA-AS3, HOXA10-AS, and HOXA11-AS; the HOXB region, consisting of HOXB-AS1, HOXB-AS2, HOXB-AS3, HOXB-AS4, and PRAC2; the HOXC region, consisting of HOTAIR, HOXC13-AS, HOXC-AS1, HOXC-AS2, and HOXC-AS3; and the HOXD region, consisting of HAGLR and HOXD-AS2. These genes showed a positive correlation with the expression of CTLA-4 and PD-L1, as well as other immunocyte markers and immune cells, such as resting T cells, NK cells, eosinophils, and mast cells [129].

In particular, HOTTIP enhances IL-6 production and suppresses T cell activity, potentiating immunological escape of ovarian cancer cells. According to one study, HOTTIP may affect ovarian cancer development and progression by modulating neutrophil activity [130]. It is responsible for activating multiple *HOXA* genes by recruiting the WRD5/MLL histone methyltransferase complex, which then trimethylates H3K4 at the *HOXA* promoter, facilitating gene activation [131]. In cancer cells, HOTTIP was observed to bind to c-jun, inducing IL-6 expression [130]. The IL-6 released to the tumor microenvironment activated IL-6/JAK/STAT3/PD-L1 in neutrophils. Several studies are testing targeting IL-6 in combination with ICIs, such as the NCT04940299 clinical trial [132]. This supports future studies in HOTTIP blockers to reduce IL-6 levels and improve ICIs benefits.

Another example related to this axis of regulation is the small nucleolar host gene 12 (SNHG12) lncRNA. SNHG12 recruits NF-κB to the IL-6R promoter, increasing IL-6R expression and promoting IL-6/miR-21 interaction between cancer cells and M2 macrophages, enhancing PD-L1 expression. This mechanism has been observed in both ovarian cancer cells and macrophages, where increased PD-L1 expression induced by SNHG12 inhibits T cell proliferation [133], thus contributing to tumor evasion.

HOX transcript antisense RNA (HOTAIR) from the HOXC cluster acts as a scaffold to gather the histone methyltransferase PRC2 and the histone demethylase LSD1, leading to chromatin remodeling that alters the expression of certain genes, such as repression of the *HOXD* gene [134]. Another mechanism studied for HOTAIR is its ability to sequester miR-30a-5p, which targets glucose regulatory protein 78 (GRP78), a major endoplasmic reticulum (ER) stress responding protein, which in turn interacts with PD-L1, increasing its stability. As a result, an increase in HOTAIR (seen in various types of cancer) results in increased PD-L1, explaining HOTAIR’s oncogenic nature [135].

In another example of the role of HOX lncRNAs, Wang et al. confirmed the existence of a direct interaction between lncRNA HOXA-AS2 and miR-519, which increases hypoxia-inducible factor-1α (HIF-1α) and PD-L1 expression [136]. Hypoxia is a common player in the tumor microenvironment. In hypoxic conditions, tumor cells activate the HIF-1α transcription factor, which binds directly to hypoxia response elements in the PD-L1 proximal promoter and upregulates their expression. NUTM2A-AS1 is another lncRNA that modulates PD-L1 through HIF-1α by sponging miR-376a in gastric cancer [137].

Xian et al. [138] reported that colorectal cancer cells secrete exosomes enriched with the lncRNA KCNQ1OT1. These exosomes can affect T cells by decreasing miR-30a-5p levels, targeting ubiquitin-specific peptidase 22 (USP22), a deubiquitinating enzyme that can deubiquitinate PD-L1 and increase its stability. Acting at another level, KCNQ1OT1 sponges miRNAs that directly target the 3′-UTR of PD-L1, such as miR-15a in prostate cancer cells [139] and miR-506 in hepatocellular carcinoma cells [140].

lncRNA LINC00473 is also strongly induced and associated with negative prognosis in pancreatic cancer. Like MALAT1, LINC00473 promotes PD-L1 expression by sponging miR-195-5p. When LINC00473 is suppressed or miR-195-5p is upregulated, PD-L1 is downregulated, which increases CD8+ T cells and inhibits cancer progression [79].

The hypoxia-inducible lncRNA nuclear-enriched autosomal transcript 1 (NEAT1) is significantly overexpressed in tumor-infiltrating T cells in hepatocellular carcinoma tumors and has been found to sequester miR-155. Yan et al. found that NEAT1 increases TIM-3 expression through modulating miR-155 expression. NEAT1 downregulation can decrease CD8+ T cell death and increases cytolysis activity against cancer [141]. In summary, strategies based on the downregulation of NEAT1, using for example targeted antisense oligonucleotide (ASO) to block NEAT1 polyadenylation [142], can open up new avenues in cancer treatment. 

The lncRNA MIAT negatively regulated the expression of miR-411-5p, an oncogenic miRNA that blocks STAT3 and modulates PD-L1 expression in hepatocellular carcinoma [143]. The miR-141/STAT3 axis has also been found in other types of cancer such as prostate and cervical cancer [144].

The lncRNAs NKX2-1-AS1—located in the 14q13.3 chromosome region—and NKX2-1 protein (also known as thyroid transcription factor 1, TTF-1) are co-expressed in lung cancer, although they have distinct effects on *CD274* gene (encoding PD-L1) [145,146]. The NKX2-1 protein can activate PD-L1 expression by binding to its gene promoter. NKX2-1-AS1, in contrast, interferes with NKX2-1 protein binding to the CD274-promoter, reducing CD274 mRNA synthesis and resulting in PD-L1 downregulation. Furthermore, when the NKX2-1-AS1 is knocked down, PD-L1 expression rises [145]. As a result, high levels of NKX2-1-AS1 are required to maintain low levels of PD-L1 expression, limiting tumor cell ability to evade the immune system.

Overexpression of miR-194-5p-targeting PD-L1 inhibited migration, invasion, and proliferation; decreased pancreatic cancer development; and increased IFN production by CD8+ T cells after increasing CD8+ T cell infiltration in the tumor microenvironment [147]. According to some studies, miRNA downregulation (e.g., via loss of miR-140 and miR-340) has also been linked to an increase in PD-L1 levels in cervical cancer [148]. According to a study on hepatocellular carcinoma, the lncRNA PCED1B-AS1 (PCED1B antisense RNA 1) acts by recruiting mir-194-5p, increasing the expression of not only PD-L1, but also PD-L2. The same study revealed that PCED1B-AS1 was present in blood exosomes of patients with advanced hepatocellular carcinoma, signaling T cells or macrophages to induce immunosuppression [149].

Recently, Ma et al. found that the lncRNA lncMX1-215 inhibits PD-L1 and galectin-9 expression in head and neck squamous cell carcinoma cells by interrupting GCN5-mediated H3K27 acetylation (an activating mark), and that its overexpression dramatically lowers tumor cell growth and metastasis in vitro and in vivo [150]. Based on this finding, lncMX1–215 can be classified as a tumor suppressor, and the authors recommended lncMX1–215 as a possible treatment target [150]. The pseudogene RP11-424C20.2 is another lncRNA that regulates immune evasion through chromatin remodeling pathways. RP11-424C20.2 may act as a ceRNA in thymomas (a tumor originating from the epithelial cells of the thymus), increasing the expression of its parental gene ubiquitin-like protein containing PHD and RING finger domains 1 (*UHRF1*) via sponging miR-378a-3p, and modulating IFN-γ-mediated CLTA-4 and PD-L1 activity [151].

Small nucleolar RNA host gene 20 (SNHG20) enhances PD-L1 expression in esophageal squamous cell carcinoma via the ATM/JAK/PD-L1 pathway [152]. The mechanism by which SNHG20 activates this pathway is not completely understood; however, one possibility would be that SNHG20 directly regulates the DNA repairing activity of the enzyme ataxia telangiectasia mutated (ATM), as occurs with other lncRNAs such as HIF-1α inhibitor at translation level (HITT) [153].

The lncRNA lnc-Tim3 is shown to be increased in tumor-infiltrating CD8+ T cells in hepatocellular carcinoma. IL-2 and IFN-γ production is negatively correlated with its overexpression. Lnc-Tim3 binds directly to TIM-3 and blocks IC interaction with the chaperone Bat3, inhibiting downstream signaling of TIM-3 [154]. Furthermore, lnc-Tim3 promotes the transcriptional activation of anti-apoptotic proteins such as B-cell lymphoma 2 (Bcl-2) and mouse double-minute homolog 2 (MDM2) through p300-dependent p53 and transcription factor p65, resulting in the survival of exhausted CD8+ T cells [154].

## 5. Super-Enhancer lncRNAs

Recent studies have found that selncRNAs regulate the expression of ICs, including stimulatory and inhibitory checkpoints [157]. For example, selncRNA CCAT1-L, classified as a nuclear-retained lncRNA, enhances MYC expression in cis, which upregulates the expression of innate IC cluster of differentiation 47 (CD47) and adaptive IC PD-L1 [157,158]. Furthermore, the CCAT1/TP63/SOX2 complex binds to EGFR SE sites to promote EGFR transcription in trans, enhancing PD-L1 expression by activating PI3K/AKT and RAF/MEK/ERK signaling. SelncRNA CCAT1 might boost PD-L1 transcription by creating a selncRNA–TF complex that promotes target gene expression and stimulates downstream signaling pathways [157,159]. Another example is the selncRNA-associated IFN-γ signaling pathway, which substantially increases PD-L1 expression in melanoma cells by activating the JAK/STAT/IRF1 axis in melanoma cells [160]. 

In a recent study on B cell lymphoma, the bromodomain and extraterminal (BET) protein BRD4 stimulated selncRNA transcription, and a chromatin loop was observed between distal SE and PD-L1 tumor-specific survival. This indicates the potential involvement of selncRNAs in BRD4-mediated signaling. SelncRNA suppression mediated by BRD4 inhibitors may either boost anticancer immunity by decreasing PD-L1 expression or inhibit anticancer immunity by inactivating immune cells [161].

## 6. Circular RNAs Mediating Immune Checkpoint Regulation

Recently, circRNAs have also been demonstrated to have a regulatory role in ICs, suggesting their potential use as targets for IC treatment. Hsa_circ_0020397 was the first circRNA found able to influence PD-L1 expression, indicating that circRNAs may also play a role in tumor immune escape. In colorectal cancer, hsa_circ_0020397 was demonstrated to block the action of the onco-suppressor miR-138, via an RNA sponge effect, thus boosting PD-L1 expression. Overexpression of PD-L1 increases cancer cell viability and proliferation while inhibiting apoptosis [162]. circRNA fibroblast growth factor receptor 1 (circFGFR1), derived from FGFR1, was shown to have a high expression level in non-small cell lung cancer and was related to a poor prognosis and clinicopathological aspects. Furthermore, circFGFR1 inhibits miR-381-3p by sponging, which promotes lung cancer cell resistance to PD-1 blockers [163]. In contrast, circ-CPA4 downregulated let-7 miRNA, leading to upregulation of PD-L1 expression in non-small cell lung cancer cells. Circ-CPA4 increased PD-L1 expression at both intracellular and extracellular (exosomal PD-L1) levels in non-small cell lung carcinoma cells [164]. circ-CPA4 depletion resulted in a considerable increase in CD8+ T cell proliferation, whereas overexpression in non-small cell lung cancer cells resulted in CD8+ T cell inactivation in co-culturing systems [164]. PD-L1 expression was similarly boosted by CircCHST15. Interestingly, animals implanted with CircCHST15-silenced cancer cells exhibited a significant increase in CD8+ T lymphocytes [165]. Furthermore, Circ-KRT6C enhanced PD-L1 levels by inhibiting miR-485–3p-mediated inhibitory effects. IFN and TNF levels in cell supernatant were found to be increased in miR-485–3p-transfected cells co-cultured with peripheral blood mononuclear cells [166]. 

CircUHRF1 in plasma exosomes can suppress the functions of natural killer cells by upregulating the expression of TIM-3 via miR-449c-5p degradation and induce resistance to anti-PD1 immunotherapy in hepatocellular carcinoma patients [167]. In mice implanted subcutaneously with circUHRF1-knockdown HCCLM3 cancer cells, anti-PD1 treatment sensitivity was found to be improved [167]. Similarly, circRNA-002178 has been shown to be transported into CD8+ T cells through exosomes to stimulate PD-1 expression. CircRNA-002178 induces T cell exhaustion by increasing PD-L1 expression via sponging miR-34 [168]. CircKRT1 has been shown to promote tumor development and immune evasion in oral squamous cell carcinoma by interfering with miR-495–3p-mediated PD-L1 inhibition [169].

## 7. Conclusions and Perspectives

The important role played by the immune system in patient response to therapies treating cancer is well known. Cancer treatment based on the use of immune checkpoint inhibitors (ICIs) has revolutionized the oncology field far beyond their remarkable clinical efficacy in some patients: T cell-targeted immunomodulators are now used as single treatment or in combination with chemotherapies as first or second treatment lines for almost 50 cancer types [16]. The predominant therapeutic strategy to inhibit ICs has been based on the use of therapeutic antibodies, mainly anti-CTLA4 (e.g., ipilimumab), anti-PD1 (e.g., pembrolizumab and nivolumab), and PD1-L1 (e.g., atezolizumab and durvalumab) to unleash cytotoxic T cells in the tumor microenvironment. However, although the use of ICIs has achieved great clinical success in several cancers, a considerable proportion of patients show no benefit after using this therapeutic approach. This underscores the importance of understanding the tumor microenvironment and heterogenicity of each cancer type in each patient [170].

This review has highlighted the regulatory role of some ncRNAs in ICs, suggesting that lncRNAs could be utilized as targets for IC treatment. Several strategies based on the therapeutic manipulation of lncRNA promoters (such as the use of double-stranded DNA plasmid carrying a gene that is regulated by a specific lncRNA gene promoter) could be administered to elicit an antitumor response in solid tumors. Others, consisting of the steric inhibition of lncRNAs using CRISPR-Cas9 technology and nucleic acid-based therapies (i.e., ASOs or interference RNAs (siRNAs)) also form the basis for a battery of feasible lncRNA-targeting therapeutic strategies to be developed against ICs. The higher success rates of nucleic acid-based therapeutics provide a promising opportunity to explore lncRNAs as potential therapeutic targets upstream of ICs. The main potential negative effect of using lncRNAs as ICs regulators is mis-targeting, given that lncRNAs can have several targets. Nonetheless, this problem could be addressed by carefully designing lncRNA-competing molecules, limiting the off-target effects, or even taking advantage of interactions between different lncRNAs to alter lncRNA networks. These include circRNAs as sponges of miRNAs, miRNA–lncRNA interactions, and even the ceRNA regulation model in which lncRNAs/circRNAs act as sponges for miRNAs to indirectly regulate miRNA downstream target genes. Here, it is worth noting that circRNAs and miRNAs can potentially be used to produce polypeptide-like antigens, receptors, or other functional proteins [171], thereby reprogramming the antitumor immune response and blocking inhibitory receptors (e.g., PD-1 and PDL-1). Alternatively, specific circRNAs or miRNAs can also plausibly be used to generate new tumor microenvironments. These environments can serve as synthetic immune niches for controlled local immunomodulation and to promote systemic antitumor immunity, as an alternative mechanism to avoid tumor evasion. 

Thanks to an increased understanding of the regulatory role of miRNAs, miRNA-based therapeutics have emerged as among the most exciting and significant therapeutic breakthroughs of recent years, and several solutions are currently at different stages of clinical trials. Recently, Hu et al. revealed that patients with PD-1 inhibitor-resistant triple-negative breast cancer exhibited elevated long intergenic non-coding RNA for kinase activation (LINK-A) levels. The authors used LINK-A-locked nucleic acids to stabilize sensitized mammary gland tumors to IC blockers, with promising results [172], heralding the future development of synthetic designed RNAs specifically targeting lncRNAs to regulate cascades involved in ICs, and thereby laying the foundation for new potential treatments. Lastly, ongoing clinical trials are exploring the use of lncRNAs as biomarkers, (including CCAT1 and HOTAIR (clinicalTrials.gov)), which may thus become available in the near future to improve immunotherapy treatments for cancer patients.

## Figures and Tables

**Figure 1 cancers-15-00184-f001:**
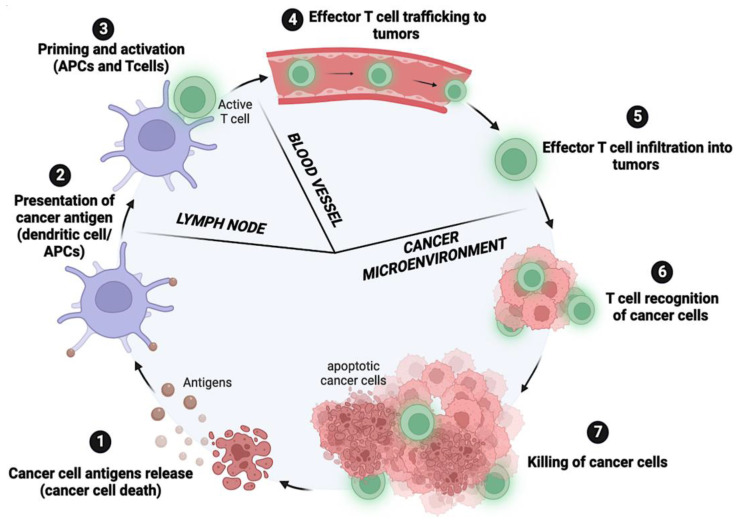
Schematic illustration of the cancer–immunity cycle, representing the seven key steps in developing an immune response against cancer: (1) Tumor antigens are released; (2) dendritic cells capture and present tumor antigens to T cells; (3) T cells are activated and differentiated to effector T cells; (4) effector T cells are trafficked to tumor cells; (5) effector T cells infiltrate the tumor tissue; (6) T cells recognize cancer cells; (7) cancer cell suppression. Adapted from [26]. Created with Bio.Render.com.

**Figure 2 cancers-15-00184-f002:**
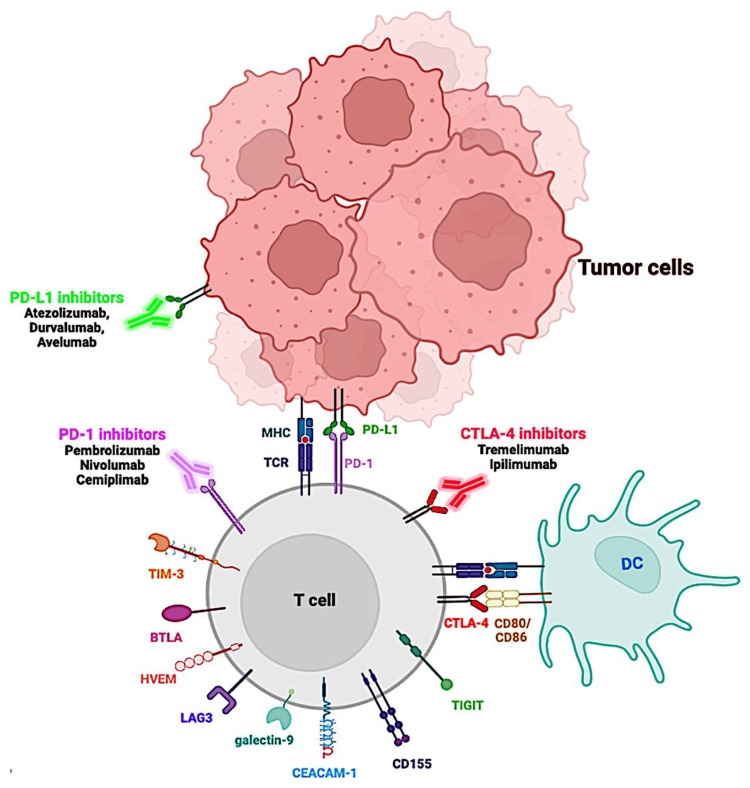
FDA-approved immune checkpoint inhibitors as a therapeutic intervention in cancer. T cells are activated through antigen presentation by major histocompatibility complex (MHC) molecules on the surface of tumor cells or on antigen-presenting cells (APC) such as dendritic cells (DC), recognized by the T cell receptor (TCR). Activated T cells will upregulate the expression of co-inhibitory cell surface receptors such as PD-1 and CTLA-4. Binding of PD-1 to its ligands, PD-L1, or PD-L2, will inhibit signaling downstream of the TCR, thus downregulating T cell activity. Antibody therapies that target PD-1 or PD-L1 can reactivate exhausted T cells at the tumor location, increasing their activity and allowing T cells to destroy tumor cells. Binding of CTLA-4 to its ligands, B7, CD80, or CD86, suppresses T cell activity. By blocking CTLA-4 interaction with anti-CTLA-4 antibodies, T cell proliferation will be activated. Created with Bio.Render.com.

**Figure 3 cancers-15-00184-f003:**
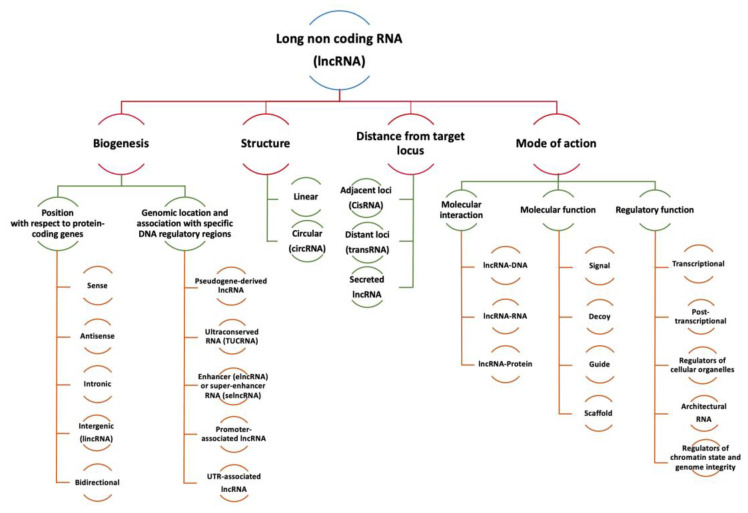
Classification of long non-coding RNAs (lncRNAs) according to biogenesis (where they are transcribed), structure, effects exerted on DNA sequence, and mode of action including molecular interaction, molecular, and regulatory function.

**Figure 4 cancers-15-00184-f004:**
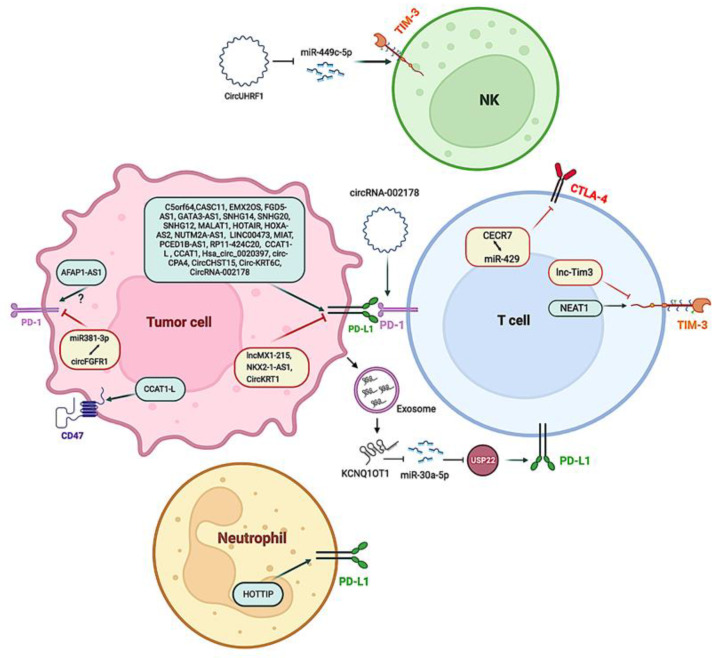
Regulation of immune checkpoint (IC) molecules by lncRNAs in human cancer. Modulatory effects of lncRNAs on T cells, neutrophils, and NK cells. lncRNAs (linear and circular RNAs) regulate different ICs, including PD-1, PD-L1, CD47, and TIM-3 in cancer cells, T cells, neutrophils, and NK cells. Green arrows denote stimulatory regulation. Red T-shaped symbols represent inhibitory regulation. Created with Bio.Render.com.

**Table 1 cancers-15-00184-t001:** The expression pattern of membrane bound isoform of immune checkpoints (ICs) and their effects on tumor microenvironment.

Immune Checkpoint	Cellular Expression	Ligand	Cellular Expression	Effects on Tumor Microenvironment	Ref.
CTLA-4 (CD152)	Tregs, Teffs, B cells, NK cells, NKT cells, and DCs	CD80 (B7-1), CD86 (B7-2)	APCs	Inhibits T cell activation by binding to its ligand. Inhibits IL-2 production and influences naive CD4+ T cell differentiation.	[29]
PD-1	Tregs, Teffs, B cells, NK cells, mast cells, and some subsets	PD-L1 (B7-H1), PD-L2 (B7-DC)	Tumor cells, non-lymphoid, and non-hematopoietic cells	Inhibits effector T cell activation and promotes Treg cell generation	[30]
LAG-3	Tregs, Teffs, B cells, NK cells, and DCs	MHC II	APCs	Has a synergistic impact with PD-1 to inhibit immune responses by suppressing T cell activation and cytokine production, thereby ensuring immune homeostasis	[31,32]
TIM-3 (HAVCR2)	Tregs, Teffs, NK cells, and some subsets of myeloid cells	Galectin-9, CEACAM1, Soluble HMGB1, PtdSer	Some myeloid subsets; Tregs, Teffs, NK cells, and some subsets of myeloid cells; Released by tumor cells or activated DCs: Apoptotic cells	Favors tumor escape to immune cells. Inhibits T cell responses. CD8+-T cells lose the ability to secrete cytokines IFNγ, IL-2, and TNFα	[33,34]
TIGIT	Tregs, Teffs, and NK cells	CD112 (PVRL2; nectin-2), CD155 (PVR)	DCs, APCs, and tumor cells	Suppresses the activation of TILs	[35,36]
BTLA (CD272)	T cells, B cells, macrophages, and NK cells	HVEM, TNFRSF14	-	Suppresses pathway for T cell, B cell, or NK.	[37,38]
IDO-1	EC, fibroblasts, macrophages, DCs, and PBMCs	GITR, ICOS, CD200	DCs	Increases intratumoral infiltration and impairs cytotoxic T cell function. In DC, decreases antigen uptake and downregulates CD40/CD80	[39]

APC: antigen-presenting cell; DC: dendritic cell; EC: endothelial cell; MHC: major histocompatibility complex; NK: natural killer cells; NKT: natural killer T cells; PMBCs: peripheral blood mononuclear cells; PtdSer: phosphatidylserine; Teff: effector T cells; Tregs: regulatory T cells; TILs: tumor-infiltrating lymphocytes.

**Table 2 cancers-15-00184-t002:** Expression pattern of soluble isoform of immune checkpoints (ICs) and their effects on tumor microenvironment.

Immune Checkpoint	Cellular Expression	Ligand	Cellular Expression	Effects in Tumor Microenvironment	Ref.
sCTLA-4	Monocytes, immature DCs, and Treg cells	CD80, CD86	APCs	Inhibits T cell responses	[56]
sPD-1	PBMCs	PD-L1/2	Tumor cells	Blocks PD-L/PD-1 interactions, Activates CD8+ T cells	[57]
sPD-L1	Mature DCs	PD-1	T cells	Combines with PD-1, inhibits T cell responses, and reduces T cell proliferation	[58,59]
sPD-L2	Activated leukocytes	PD-1	-	Unknown function	[60]
sCD80 (sB7–1)	unstimulated B cells and monocytes, and activated T and B cells	CTLA-4, CD28	T cells	Inhibits PD-1/PD-L1 pathway, T cell proliferation, and IL-2 production	[61]
sCD86 (sB7–2)	Constitutively expressed on APCs, monocytes, DC, and certain cancer cells	CTLA-4, CD28	T cells	Inhibits T cell responses	[61]
sB7-H3	Monocytes, DCs, and activated T cells	B7-H3R	T cells	Promotes IL-8 and VEGF expression, increasing invasion and metastases of pancreatic carcinoma cells	[62,63]
sCD137 (s4-1BB)	Activated PBMCs	CD137L (4-1BBL)	T cells	Inhibits CD137/CD137L pathway	[64]

APC: antigen-presenting cell; DC: dendritic cell; PMBCs: peripheral blood mononuclear cells; Tregs: regulatory T cells.

**Table 3 cancers-15-00184-t003:** LncRNAs related to immune evasion in cancer. The last column summarizes the regulatory pathways detailed in the article.

Name	Ensembl ID	Chromosome	Strand	Class	Mechanism of Action Related to Immune Evasion in Cancer
AFAP1-AS1	ENSG00000272620	4	+	Intergenic	AFAP-AS1/PAG/PD-1
C5orf64	ENSG00000178722	5	+	Intergenic	C5orf64/miRNA-150/EGFR/PD-L1
CASC11	ENSG00000249375	8	−	Intergenic	CASC11/EIF4A3/E2F1/NF-κB/PD-L1
					CASC11/EIF4A3/E2F1/PI3K/AKT/mTOR/PD-L1
CECR7	ENSG00000237438	22	+	Intergenic	CECR7/miR-429/CTLA4
EMX2OS	ENSG00000229847	10	−	Antisense	EMXOS/miR-654-3p/AKT3/PD-L1
FGD5-AS1	ENSG00000225733	3	−	Antisense	FGD5-AS1/miR-454-3p/ZEB1/PD-L1
					FGD5-AS1/miR-142/PD-L1
GATA3-AS1	ENSG00000197308	10	−	Antisense	GATA3-AS1/miR-676-5p/COPS5/PD-L1 stability
HOTAIR	ENSG00000228630	12	−	Intergenic	HOTAIR/miR-30a-5p/GRP78/PD-L1 stability
HOTTIP	ENSG00000243766	7	+	Antisense	HOTTIP/c-jun/IL-6
HOXA-AS2	ENSG00000253552	7	+	Antisense	HOXA-AS2/miR-519/PD-L1
					HOXA-AS2/miR-519/HIF-1a/PD-L1
KCNQ1OT1	ENSG00000269821	11	−	Antisense	KCNQ1OT1/miR-30a-5p/USP22/PD-L1 stability
					KCNQ1OT1/miR-15a/PD-L1
					KCNQ1OT1/miR-506/PD-L1
LINC00473	ENSG00000223414	6	−	Intronic	LINC00473/miR-195-5p/PD-L1
Lnc-Tim3	ENST00000443947.1	7	+	Intergenic	lnc-Tim3/TIM3 blocking
LncMX1–215					lncMX1–215/GCN5/H3K27ac in PD-L1 and Galectin-9
MALAT1	ENSG00000251562	11	+	Intergenic	MALAT1/miR-200c/ZEB1/PD-L1
					MALAT1/mir-200a/PD-L1 stability
					MALAT1/miR-195/PD-L1 stability
MIAT	ENSG00000225783	22	+	Intergenic	MIAT/miR-411-5p/STAT3/PD-L1
NEAT1	ENSG00000245532	11	+	Intergenic	NEAT1/miR-155/TIM3
NKX2-1-AS1	ENSG00000253563	14	+	Pseudogene	NKX2-1-AS1/NKX2-1/PD-L1
NUTM2A-AS1	ENSG00000223482	10	−	Intergenic	NUTM2A-AS1/miR-376a/PD-L1
PCED1B-AS1	ENSG00000247774	12	−	Antisense	PCED1B-AS1/miR-149-5p/PD-L1 and PD-L2
SNHG12	ENSG00000197989	1	−	Antisense	SNHG20/IL-6R/IL6
SNHG14	ENSG00000224078	15	+	Antisense	SNHG14/miR-5590-3p/ZEB1/PD-L1

Ensembl gene annotation, chromosome, strand, and class of each lncRNA were obtained from the LncRNA spatial atlas V2.0 (http://bio-bigdata.hrbmu.edu.cn/LncSpA, accessed on 12 July 2022) [155], NPInter v4.0 (http://bigdata.ibp.ac.cn, accessed on 12 July 2022) [156], and Lncipedia v5.2 (https://lncipedia.org, accessed on 12 July 2022).

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
