# Peer review of "Long Non-Coding RNAs as Epigenetic Regulators of Immune Checkpoints in Cancer Immunity"

_cancers, 2022, doi:10.3390/cancers15010184_

Round 1
Reviewer 1 Report
The authors present a review about the impact of non-coding RNAs in immunotherapy, particularly in immune check point inhibitors. The review is opportune and covers a growing area of interest.
In the manuscript, authors focus on Long non-coding RNAs, which are less well known than micro RNAs, making the article of interest for a wide audience. The review provides an extensive description of LncRNAs and provides a very interesting proposal to classify LncRNAs according to their structure, function, etc. I have found this classification of particular interest as it could clarify future LncRNAs nomenclature and assist authors. Other point that I find interesting is how authors describe and review the mechanistic aspects involving LncRNA modulation of the response to immunotherapy. Perhaps, the same level of detail should be provided in the circRNA and seRNA section
Overall, I find the article interesting and it compiles the most recent published information. I would like to highlight the information and the quality of the figures.
Author Response
Thanks a lot for your comments. Major changes have been marked in red to facilitate their identification.
The authors present a review about the impact of non-coding RNAs in immunotherapy, particularly in immune check point inhibitors. The review is opportune and covers a growing area of interest.
In the manuscript, authors focus on Long non-coding RNAs, which are less well known than micro RNAs, making the article of interest for a wide audience. The review provides an extensive description of LncRNAs and provides a very interesting proposal to classify LncRNAs according to their structure, function, etc. I have found this classification of particular interest as it could clarify future LncRNAs nomenclature and assist authors. Other point that I find interesting is how authors describe and review the mechanistic aspects involving LncRNA modulation of the response to immunotherapy. Perhaps, the same level of detail should be provided in the circRNA and seRNA section
ANSWER: We would like to thank the reviewer for considering our manuscript opportune and interesting for a wide audience. Your comment regarding to present in this manuscript more information about the role of circRNAs and seRNAs in modulating the response to immunotherapy is very appropriate. Unfortunately, we have not found enough information in the current literature to provide further information in this section.
Overall, I find the article interesting and it compiles the most recent published information. I would like to highlight the information and the quality of the figures.
ANSWER: We thank the reviewer for the positive comments regarding the content of the manuscript and the quality of the figures.

Reviewer 2 Report
Review- CANCERS-2064179
Long Non-coding RNAs as epigenetic regulators of immune checkpoints in cancer immunity
Wiam Saadi, et al.
General comments:
Writing is good, but simple checks for grammar needed.
The authors’ goal is to highlight lncRNA candidates that are more likely to have a therapeutic impact in the near future and might be innovative cancer treatment targets. The review captured research from various publications but throughout the body of the text, very little is offered in regard to the authors’ perspectives on each of the subtopics reviewed.
The conclusions and perspectives section is very short. This raises the question of what this paper contributes to the literature. Not much is highlighted by way of the writing style. The authors need to add more perspective putting their findings in text throughout the paper. Otherwise, they have offered nothing new to the research community interested in cancer immunity and ncRNAs.
Simple Summary:
Among “its” many functions…
What is the “it” you are referring you? Your previous sentence talks about ncRNA’s as a collective. Should “it” be “their”? Please update the sentence to be clearer.
Similar comments regarding “them” in the sentence indicated below.
… some of the ncRNAs allow “them”… What “them” are you referring to? If it is ncRNA
Abstract:
Please correct the tense in the following sentence.
… lncRNAs that have a well-described interactions… It should either be “a well-described interaction” OR “well-described interactions”
Also it would be great if the abstract and the paper in general reflect a greater need for a review type article. Your article isn’t adding to the scientific knowledge so why should it be published? Have papers reflected varied conclusions, are publications sparse and difficult to find for comparison, is there a need to summarize findings to provide better direction to researcher. The authors need to state what gap this review fills.
Introduction:
Pg 2; ln 53: Should it read epigenetic markers? It currently say epigenetic marks.
Conclusions and Perspectives:
Pg 17; ln 561 Please correct the sentence “A generally know the immune system…”
Author Response
Thank you very much for your careful review and comments. Major changes have been marked in red to facilitate their identification.
General comments:
Writing is good, but simple checks for grammar needed.
The authors’ goal is to highlight lncRNA candidates that are more likely to have a therapeutic impact in the near future and might be innovative cancer treatment targets. The review captured research from various publications but throughout the body of the text, very little is offered in regard to the authors’ perspectives on each of the subtopics reviewed.
The conclusions and perspectives section is very short. This raises the question of what this paper contributes to the literature. Not much is highlighted by way of the writing style. The authors need to add more perspective putting their findings in text throughout the paper. Otherwise, they have offered nothing new to the research community interested in cancer immunity and ncRNAs.
ANSWER: We thank the reviewer for its recommendation of adding more perspective issues with the aim of adding new knowledge to the research community. We hope this new version will satisfy the requirements of the reviewer.
Simple Summary:
Among “its” many functions…
What is the “it” you are referring you? Your previous sentence talks about ncRNA’s as a collective. Should “it” be “their”? Please update the sentence to be clearer.
ANSWER: The manuscript has been revised by a native English speaker to correct any grammar mistake.
Similar comments regarding “them” in the sentence indicated below.
… some of the ncRNAs allow “them”… What “them” are you referring to? If it is ncRNA
ANSWER: The sentence has been corrected
Abstract:
Please correct the tense in the following sentence.
… lncRNAs that have a well-described interactions… It should either be “a well-described interaction” OR “well-described interactions”
ANSWER: The manuscript has been extensively revised by an native English speaker to correct all grammar mistakes.
Also it would be great if the abstract and the paper in general reflect a greater need for a review type article. Your article isn’t adding to the scientific knowledge so why should it be published? Have papers reflected varied conclusions, are publications sparse and difficult to find for comparison, is there a need to summarize findings to provide better direction to researcher. The authors need to state what gap this review fills.
ANSWER: We thank the reviewer for his/her criticism, and we consider comments very constructive to improve the quality of our work. Following his/her suggestion, we have proceeded to discuss current strategies to modulate ICs and also provide our point of view of such strategies, even proposing some ideas on how these strategies can further be developed. So, we have rewritten some of the content to include further knowledge, including constructive criticism of the works described and the necessity of new approaches to elucidate the intricate mechanisms of interaction of lncRNAs and key immune checkpoints that contribute to tumor evasion. Moreover, we have better explained the urgent need to develop new methods to specifically target lncRNAs regulating complex cascades involved in the ICs, which may help to set the bases for new personalized treatments. In this regard, we also proposed in this review feasible mechanisms of action for lncRNAs targeting immune checkpoint that may contribute to the comprehension of tumor evasion and discussed how the use of noncoding RNA therapies could be used to generate new tumor microenvironments to produce synthetic immune niches for controlled local immunomodulation, promote systemic antitumour immunity to finally avoid tumor evasion.
All these aspects have been included in the abstract and in the main text to provide new knowledge to the state of art.
Introduction:
Pg 2; ln 53: Should it read epigenetic markers? It currently say epigenetic marks.
ANSWER: The sentence has been corrected
Conclusions and Perspectives:
Pg 17; ln 561 Please correct the sentence “A generally know the immune system…”
ANSWER: This section has been completely rewritten following the suggestions and comments raised by the reviewers. Particularly, the sentence indicated by the reviewer has been written as follows: “The important role played by the immune system in patient response to therapies treating cancer is well known”.
Reviewer 3 Report
In the present review, Saadi and colleagues delve into the mechanisms involved in the regulation of immune checkpoints (ICs). They focus in particular on the role played by several non-coding RNA (ncRNA) families, specifically long non-coding RNAs (lncRNA), super enhancer lncRNAs (selncRNAs) and circular RNAs (circRNAs).
Globally, I consider this review of interest to a broad audience considering the intricate regulatory networks were the examined ncRNAs are involved, targeting different molecular pathways; eventually, as the Authors anticipate, methods to specifically target lncRNAs regulating complex cascades involved in the ICs will realistically be developed in the coming years, setting the bases for new personalized treatments.
This review, however, has some aspects that need to be fixed before it can be recommended for publication.
MAJOR POINTS
The review needs a deep checking of the English language, e.g. verbs that are actually not conjugated in a proper way; shift from singular to plural.
Line 64: ‘Immune checkpoints (ICs) are specific membrane molecules’ would need a proper reference.
Lines 198-199: ‘Increased PD-L1 and LAG-3 expression…’, here the introduction of "next gen" ICs is a bit too abrupt. Moreover, similarly to BTLA, it would be nice to have a brief overview of the nature/role of each next-gen member.
Lines 209-214: is it possible to slightly expand this section on soluble immune checkpoints?
Line 241: references [64, 65] seem to be wrong.
Lines 252-253: the sentence ‘intergenic lncRNAs (lincRNAs) when they are located entirely in intronic regions of a PCG without any PCG intersection’ is wrong, this is not a definition of ‘intergenic’.
Lines 258-269: it would be useful to have some references for every group.
Line 269: reference [83] seems to be wrong.
MINOR POINTS
I would also recommend applying the following changes:
Line 16: replace ‘system’ with ‘response’; remove ‘to develop therapies’, the meaning is clear and it would avoid repeating ‘therapies’ too often.
Lines 22 and 572: replace ‘tumoral’ with ‘tumor’.
Line 25: replace ‘it’ with ‘they’.
Line 26: it’s ‘interactions’ (plural).
Lines 26-27: this statement is too general, since it would lead to include basically all the lncRNAs with a known interacting partner, which is not the case.
Line 41: replace ‘While’ with ‘Since’.
Line 106: ‘cells coexist. And a last escape phase, …’ the two sentences should be merged (i.e. remove the full stop)
Line 143: replace ‘expression’ with ‘expression pattern’ or something similar, I think ‘expression’ alone is a bit reductive. Replace ‘form’ with ‘forms’.
Table 1, ‘BTLA’ entry, replace ‘macrophage’ with ‘macrophages’. I would also recommend splitting the table in two (membrane and soluble isoforms separately): maybe use arrows instead of the words "increase" and "decrease", in order to have the first table on one page. Also try to add the header to the second table, to make it easier to understand?
Lines 158-159: ‘in cancer to indicate only PD1 and CTLA-4’, it is not very clear what the Authors mean here.
Line 193: here, the Authors should briefly introduce the concept of "next generation".
Line 195: replace ‘Increased of ICs expression’ with ‘Increased ICs expression’ OR ‘Increase of ICs expression’, or simply use ‘Overexpression of ICs’.
Line 202: ‘BTLS is a’ (add ‘a’).
Line 207: ‘it induces’ (add ‘it’).
Line 217: ‘of the human genome’ (add ‘the’).
Lines 221-223: I think also snoRNAs should be added?
Lines 226-226: ‘govern and regulate’, without ‘s’.
Line 230: I would suggest replacing ‘Moreover’ with ‘For these reasons’.
Line 233: replace ‘in all’ with ‘in several’.
Lines 236-239: I would recommend citing and using the updated data from the FANTOM6 project (doi: 10.1038/nature21374) instead of reference [75]. The same applies also for the following two citations, which are quite dated.
Line 251: replace ‘if are located’ with ‘if located’.
Line 254: replace ‘when are’ with ‘when they are’.
Line 268: there is a point (6) but there is no (5).
Line 280: replace ‘Other’ with ‘Another’.
Line 302: replace ‘when regulate’ with ‘when they regulate the’.
Figure 3: ‘with respect to’ (or ‘in respect of’); ‘inTronic’; choose singular or plural and stick to it [Biogenesis section]. Maybe use something like ‘Distance from target locus’ instead of ‘Effects exerted on DNA sequence’? Write ‘Scaffold’ with the upper case in the ‘Mode of action’ section.
Line 317: replace ‘associated to’ with ‘associated with’.
Line 318: I recommend writing ‘Tang and colleagues.
Line 320: replace ‘levels was’ with ‘levels were’.
Lines 324-325: the link for NPInter seems to be broken. Moreover, please also provide a brief explanation on what NPInter is.
Line 331: replace ‘has found’ with ‘was found’.
Line 333: replace ‘pathways as’ with ‘pathways such as’. The same at line 335, ‘Different TFs as’.
Line 336: replace ‘bind to CD274 promoter’ with ‘bind to the CD274 promoter’. Besides, Authors only specify at line 452 that this is the gene encoding the PD-L1 protein. This should be better explained here.
Lines 337-338: use ‘an upstream effector’, or just ‘upstream of’.
Lines 338-339: the end of this paragraph is a bit rushed and would need some rephrasing.
Line 341: which mature sequence of mir-150, -5p? Please specify, since you talk about -3p a few lines below.
Line 350: ‘of the (E2F’, please delete ‘(‘.
Lines 351-352: ‘pathwayS’.
Line 356: I would call it 'metastasized colorectal cancer'.
Line 373: ‘regulateS’.
Line 391: ‘cancer cell lines’ (not ‘cells’).
Line 425: ‘interaction’ (singular).
Line 427: ‘cells activate the’.
Line 437: replace ‘associated to’ with ‘associated with’.
Line 441: replace ‘significantly expressed’ with ‘significantly overexpressed’ (or ‘upregulated’).
Line 443: ‘increaseS’.
Line 495: ‘pathwayS’.
Line 497: replace ‘has been’ with ‘were’.
Line 502: ‘regulate’.
Lines 513-516: use ‘selncRNAs’ (plural).
Line 557: use ‘arrows denote’ (plural) since you do the same for the red T-shaped symbols.
Line 561: ‘known’.
Line 575: replace ‘it’ with ‘they’ and ‘a target’ with ‘targets’.
Line 583: replace ‘one’ with ‘some’ (or turn everything into singular).
Line 585: ‘patients with’.
Author Response
We would like to thank the reviewer for his/her detailed correction. All changes have been marked in red to facilitate their identification. However, we want to indicate that a Native English Speaker has revised the manuscript, so some of changes cannot be identified because the complete correction of the sentence.
In the present review, Saadi and colleagues delve into the mechanisms involved in the regulation of immune checkpoints (ICs). They focus in particular on the role played by several non-coding RNA (ncRNA) families, specifically long non-coding RNAs (lncRNA), super enhancer lncRNAs (selncRNAs) and circular RNAs (circRNAs).
Globally, I consider this review of interest to a broad audience considering the intricate regulatory networks were the examined ncRNAs are involved, targeting different molecular pathways; eventually, as the Authors anticipate, methods to specifically target lncRNAs regulating complex cascades involved in the ICs will realistically be developed in the coming years, setting the bases for new personalized treatments.
This review, however, has some aspects that need to be fixed before it can be recommended for publication.
ANSWER: Thank you very much for consider our manuscript of interest and your recommendations to improve the content of our manuscript. We have included all of your corrections.
MAJOR POINTS
The review needs a deep checking of the English language, e.g. verbs that are actually not conjugated in a proper way; shift from singular to plural.
ANSWER: Thank you very much for your comment. We have proceeded to submit this new version of our manuscript to a English Editing Language service.
Line 64: ‘Immune checkpoints (ICs) are specific membrane molecules’ would need a proper reference.
ANSWER: We have included the reference: Pardoll, D. The blockade of immune checkpoints in cancer immunotherapy. Nat Rev Cancer 12, 252–264 (2012). https://doi.org/10.1038/nrc3239
Lines 198-199: ‘Increased PD-L1 and LAG-3 expression…’, here the introduction of "next gen" ICs is a bit too abrupt. Moreover, similarly to BTLA, it would be nice to have a brief overview of the nature/role of each next-gen member.
ANSWER: We have revised this paragraph, the new paragraph reads as follows “Targeting of PD-1 and CTLA-4 formed the basis of the first generation of checkpoints, with the next generation including LAG-3, TIM-3, BTLA and TIGIT. Like PD-1 and CTLA, these ligands regulate immune suppression by inhibiting T cell activation and cytokine secretion. This data is shown in more detail in Table 1.”
Lines 209-214: is it possible to slightly expand this section on soluble immune checkpoints?
ANSWER: According your suggestion, we have better explained the role of soluble forms introducing this sentence in the new version of the manuscript. In this regard we have included this information “Interestingly, ICs molecules also have soluble isoforms, formed either by proteolytic cleavage of the membrane-bound form or by alternatively spliced mRNA translation and subsequent secretion of the protein product by immune cells. There is scientific evidence that these entities can participate in immune regulation through interactions between soluble form receptors and full-length ligands or between soluble ligands and full-length receptors”.
Line 241: references [64, 65] seem to be wrong.
ANSWER: It’s a mistake. This sentence is cited by Prensner 2011 and Guttman 2009. We have modified references 64 and 65 accordingly.
Lines 252-253: the sentence ‘intergenic lncRNAs (lincRNAs) when they are located entirely in intronic regions of a PCG without any PCG intersection’ is wrong, this is not a definition of ‘intergenic’.
ANSWER: We have corrected this sentence as follows: intergenic lncRNAs (lincRNAs) when they are located in intermediated regions of PCG.
Lines 258-269: it would be useful to have some references for every group.
ANSWER: Following your suggestion we have included 5 new references.
Line 269: reference [83] seems to be wrong.
ANSWER: We have corrected this reference and introduced the appropriate one (Aliperti 2021).
MINOR POINTS
I would also recommend applying the following changes:
Line 16: replace ‘system’ with ‘response’; remove ‘to develop therapies’, the meaning is clear and it would avoid repeating ‘therapies’ too often.
ANSWER: corrected
Lines 22 and 572: replace ‘tumoral’ with ‘tumor’.
ANSWER: corrected
Line 25: replace ‘it’ with ‘they’.
ANSWER: corrected
Line 26: it’s ‘interactions’ (plural).
ANSWER: The main manuscript and the abstract have been reviewed by a Native English Speaker.
Lines 26-27: this statement is too general, since it would lead to include basically all the lncRNAs with a known interacting partner, which is not the case.
ANSWER: We have changed the sentence
Line 41: replace ‘While’ with ‘Since’.
ANSWER: corrected
Line 106: ‘cells coexist. And a last escape phase, …’ the two sentences should be merged (i.e. remove the full stop)
ANSWER: This sentence have been rewritten as follows “The next phase is an equilibrium stage, with coexistence between tumor cells and im-mune cells. In the final phase the tumor escapes from immune control ”.
Line 143: replace ‘expression’ with ‘expression pattern’ or something similar, I think ‘expression’ alone is a bit reductive. Replace ‘form’ with ‘forms’.
ANSWER: corrected
Table 1, ‘BTLA’ entry, replace ‘macrophage’ with ‘macrophages’.
ANSWER: corrected
I would also recommend splitting the table in two (membrane and soluble isoforms separately): maybe use arrows instead of the words "increase" and "decrease", in order to have the first table on one page. Also try to add the header to the second table, to make it easier to understand?
ANSWER: Following your suggestion we have split the table. So, in the new version of the manuscript appear Table 1 and Table 2. Table 1 refers to “The expression pattern of membrane bound isoform of immune checkpoints (ICs) and their effects in tumor microenvironment” and Table 2 refers to “The expression pattern of soluble isoform of immune checkpoints (ICs) and their effects in tumor microenvironment.”
Lines 158-159: ‘in cancer to indicate only PD1 and CTLA-4’, it is not very clear what the Authors mean here.
ANSWER: We have better explained this sentence in order to clarify its meaning.
Line 193: here, the Authors should briefly introduce the concept of "next generation".
ANSWER: We have included a brief explanation of “next generation”
Line 195: replace ‘Increased of ICs expression’ with ‘Increased ICs expression’ OR ‘Increase of ICs expression’, or simply use ‘Overexpression of ICs’.
ANSWER: corrected
Line 202: ‘BTLS is a’ (add ‘a’).
ANSWER: We have modified the sentence as follows: “BTLA (B- and T-lymphocyte attenuator) is a receptors expressed on the surface of T cells, B cells, DC and myeloid cells with significantly higher expression in cancer patients compared to healthy controls”
Line 207: ‘it induces’ (add ‘it’).
ANSWER: corrected
Line 217: ‘of the human genome’ (add ‘the’).
ANSWER: corrected
Lines 221-223: I think also snoRNAs should be added?
ANSWER: Ok. SnoRNAs have been included.
Lines 226-226: ‘govern and regulate’, without ‘s’.
ANSWER: corrected
Line 230: I would suggest replacing ‘Moreover’ with ‘For these reasons’.
ANSWER: corrected
Line 233: replace ‘in all’ with ‘in several’.
ANSWER: corrected
Lines 236-239: I would recommend citing and using the updated data from the FANTOM6 project (doi: 10.1038/nature21374) instead of reference [75]. The same applies also for the following two citations, which are quite dated.
ANSWER: We have edited this paragraph to clarify the message and we have revised the data, including the current GENCODE v42 data.
Line 251: replace ‘if are located’ with ‘if located’.
ANSWER: corrected
Line 254: replace ‘when are’ with ‘when they are’.
ANSWER: corrected
Line 268: there is a point (6) but there is no (5).
ANSWER: corrected
Line 280: replace ‘Other’ with ‘Another’.
ANSWER: corrected
Line 302: replace ‘when regulate’ with ‘when they regulate the’.
ANSWER: corrected
Figure 3: ‘with respect to’ (or ‘in respect of’); ‘inTronic’; choose singular or plural and stick to it [Biogenesis section]. Maybe use something like ‘Distance from target locus’ instead of ‘Effects exerted on DNA sequence’? Write ‘Scaffold’ with the upper case in the ‘Mode of action’ section.
ANSWER: we have included all these corrections following your indications.
Line 317: replace ‘associated to’ with ‘associated with’.
ANSWER: corrected
Line 318: I recommend writing ‘Tang and colleagues.
ANSWER: corrected
Line 320: replace ‘levels was’ with ‘levels were’.
ANSWER: corrected
Lines 324-325: the link for NPInter seems to be broken. Moreover, please also provide a brief explanation on what NPInter is.
ANSWER: We have included a brief explanation of NPInter
Line 331: replace ‘has found’ with ‘was found’.
ANSWER: corrected
Line 333: replace ‘pathways as’ with ‘pathways such as’. The same at line 335, ‘Different TFs as’.
ANSWER: corrected
Line 336: replace ‘bind to CD274 promoter’ with ‘bind to the CD274 promoter’. Besides, Authors only specify at line 452 that this is the gene encoding the PD-L1 protein. This should be better explained here.
ANSWER: We have revised this sentence to make it clearer.
Lines 337-338: use ‘an upstream effector’, or just ‘upstream of’.
ANSWER: corrected
Lines 338-339: the end of this paragraph is a bit rushed and would need some rephrasing.
ANSWER: We have rephrased the sentence.
Line 341: which mature sequence of mir-150, -5p? Please specify, since you talk about -3p a few lines below.
ANSWER: We have changed by: MiR-150 also targets EREG or epiregulin [both mi-150-5p [104], and mir-150-3p (Tar-getScan; https://www.targetscan.org)]
Line 350: ‘of the (E2F’, please delete ‘(‘.
ANSWER: the sentence has been properly corrected as follows “the complex increases the stability of the E2F Transcription Factor 1 (E2F1) mRNA”
Lines 351-352: ‘pathwayS’.
ANSWER: corrected
Line 356: I would call it 'metastasized colorectal cancer'.
ANSWER: corrected
Line 373: ‘regulateS’.
ANSWER: corrected
Line 391: ‘cancer cell lines’ (not ‘cells’).
ANSWER: corrected
Line 425: ‘interaction’ (singular).
ANSWER: corrected
Line 427: ‘cells activate the’.
ANSWER: corrected
Line 437: replace ‘associated to’ with ‘associated with’.
ANSWER: corrected
Line 441: replace ‘significantly expressed’ with ‘significantly overexpressed’ (or ‘upregulated’).
ANSWER: corrected
Line 443: ‘increaseS’.
ANSWER: corrected
Line 495: ‘pathwayS’.
ANSWER: corrected
Line 497: replace ‘has been’ with ‘were’.
ANSWER: corrected
Line 502: ‘regulate’.
ANSWER: corrected
Lines 513-516: use ‘selncRNAs’ (plural).
ANSWER: corrected
Line 557: use ‘arrows denote’ (plural) since you do the same for the red T-shaped symbols.
ANSWER: corrected
Line 561: ‘known’.
ANSWER: corrected
Line 575: replace ‘it’ with ‘they’ and ‘a target’ with ‘targets’.
ANSWER: corrected
Line 583: replace ‘one’ with ‘some’ (or turn everything into singular).
ANSWER: corrected
Line 585: ‘patients with’.
ANSWER: corrected
Round 2
Reviewer 3 Report
I would like to thank the Authors for taking the time to read my suggestion and to fully implement the recommended changes. English has also been significantly improved and this very interesting review is now much easier to read and understand.